# Learning Latent Representations for Inverse Dynamics using Generalized Experiences

## Abstract

Many practical robot locomotion tasks require agents to use control policies that can be parameterized by goals. Popular deep reinforcement learning approaches in this direction involve learning goal-conditioned policies or value functions, or Inverse Dynamics Models (IDMs). IDMs map an agent's current state and desired goal to the required actions. We show that the key to achieving good performance with IDMs lies in learning the information shared between equivalent experiences, so that they can be generalized to unseen scenarios. We design a training process that guides the learning of latent representations to encode this shared information. Using a limited number of environment interactions, our agent is able to efficiently navigate to arbitrary points in the goal space. We demonstrate the effectiveness of our approach in high-dimensional locomotion environments such as the Mujoco Ant, PyBullet Humanoid, and PyBullet Minitaur. We provide quantitative and qualitative results to show that our method clearly outperforms competing baseline approaches.

## 1 Introduction

In reinforcement learning (RL), an agent optimizes its behaviour to maximize a specific reward function that encodes tasks such as moving forward or reaching a target. After training, the agent simply executes the learned policy from its initial state until termination. In practical settings in robotics, however, control policies are invoked at the lowest level of a larger system by higher-level components such as perception and planning units. In such systems, agents have to follow a dynamic sequence of intermediate waypoints, instead of following a single policy until the goal is achieved. A typical approach to achieving goal-directed motion using RL involves learning goal-conditioned policies or value functions (Schaul et al. (2015)). The key idea is to learn a function conditioned on a combination of the state and goal by sampling goals during the training process. However, this approach requires a large number of training samples, and does not leverage waypoints provided by efficient planning algorithms. Thus, it is desirable to learn models that can compute actions to transition effectively between waypoints. A popular class of such models is called *Inverse Dynamics Model* (IDM) (Christiano et al. (2016); Pathak et al. (2017)). IDMs typically map the current state (or a history of states and actions) and the goal state, to the action.

In this paper, we address the need of an efficient control module by learning a *generalized IDM* that can achieve goal-direction motion by leveraging data collected while training a state-of-the-art RL algorithm. We do not require full information of the goal state, or a history of previous states to learn the IDM. We learn on a reduced goal space, such as 3-D positions to which the agent must learn to navigate. Thus, given just the intermediate 3-D positions, or waypoints, our agent can navigate to the goal, without requiring any additional information about the intermediate states. The basic framework of the IDM is shown in Fig. 1.

The unique aspect of our algorithm is that we eliminate the need to randomly sample goals during training. Instead, we exploit the known symmetries/equivalences of the system (as is common in many robotics settings) to guide the collection of *generalized experiences* during training. We propose a class of algorithms that utilize the property of equivalence between transitions modulo the difference in a fixed set of attributes. In the locomotion setting, the agent's transitions are symmetric under translations and rotations. We capture this symmetry by defining *equivalence modulo orientation* among experiences. We use this notion of equivalence to guide the training of latent

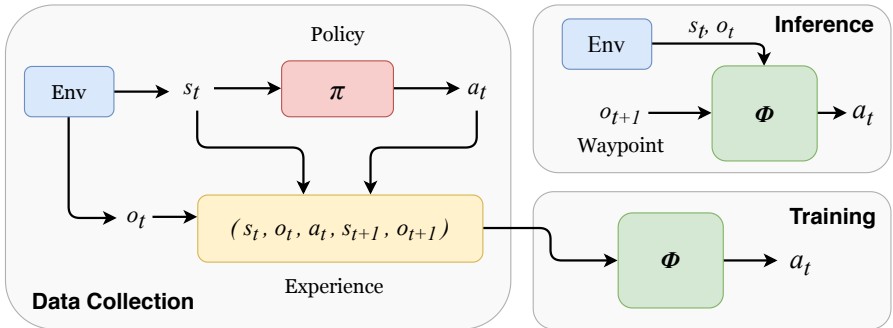

Figure 1: Basic framework of the Inverse Dynamics Model. Experiences are collected in normal training under single-goal reward function and used to train the IDM to produce actions for the agent to move from $o_t$ to $o_{t+1}$. At inference time, the current state $s_t$ and position $o_t$ are passed to the IDM, along with immediate next goal/waypoint $o_{t+1}$ to get the required action $a_t$.

representations shared by these experiences and provide them as input to the IDM to produce the desired actions, as shown in Fig. 4. A common challenge faced by agents trained using RL techniques is lack of generalization capability. The standard way of training produces policies that work very well on the states encountered by the agent during training, but often fail on unseen states. Achieving good performance using IDMs requires both these components: collecting generalized experiences, and learning these latent representations, as we demonstrate in Section 6. Our model exhibits high sample efficiency and superior performance, in comparison to other methods involving sampling goals during training.

We demonstrate the effectiveness of our approach in the Mujoco Ant environment (Todorov et al. (2012)) in OpenAI Gym (Brockman et al. (2016)), and the Minitaur and Humanoid environments in PyBullet (Coumans & Bai (2016)). From a limited number of experiences collected during training under a single reward function of going in one direction, our generalized IDM succeeds at navigating to arbitrary goal positions in the 3-D space. We measure performance by calculating the closest distance to the goal an agent achieves. We perform ablation experiments to show that (1) collecting generalized experience in the form of equivalent input pairs boosts performance over all baselines, (2) these equivalent input pairs can be condensed into a latent representation that encodes relevant information, and (3) learning this latent representation is in fact critical to success of our algorithm. Details of experiments and analysis of results can be found in Sections 5 and 6.

## 2 RELATED WORK

Several recent works learn policies and value functions that are conditioned over not just the state space, but also the goal space (Andrychowicz et al. (2017); Schaul et al. (2015); Kulkarni et al. (2016)) and then generalize those functions to unseen goals. Goal-conditioned value functions are also largely used in hierarchical reinforcement learning algorithms (Kulkarni et al. (2016)), where the higher level module learns over intrinsic goals and the lower level control modules learn sub-policies to reach those goals, or the lower level control modules can efficiently execute goals proposed by the higher-level policy (Nachum et al. (2018)). Ghosh et al. (2018) use trained goal-conditioned policies to learn actionable latent representations that extract relevant information from the state, and use these pretrained representations to train the agent to excel at other tasks. Pong* et al. (2018) learn goal-conditioned value functions, and use them in a model-based control setting.

IDMs are functions that typically map the current state of the agent and the goal state that the agent aims to achieve, to the desired action. They have been used in a wide variety of contexts in existing literature. Christiano et al. (2016) train IDMs using a history of states and actions and full goal state information for transferring models trained in simulation to real robots. Pathak et al. (2017) and Agrawal et al. (2016) use IDMs in combination with Forward Dynamics Models (FDMs) to predict actions from compressed representations of high-dimensional inputs like RGB images generated by the FDM. Specifically, Pathak et al. (2017) use IDMs to provide a curiosity-based reward signal

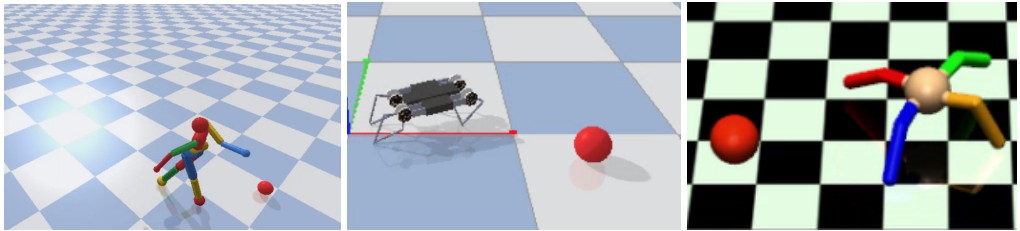

Figure 2: We report the results of our algorithm and baselines on the three locomotion environments shown above: Humanoid, Minitaur, and Ant. The red sphere indicates the goal in each environment.

in the general RL framework to encourage exploration; Agrawal et al. (2016) use IDMs to provide supervision for the learning of visual features relevant to the task assigned to the robot.

We circumvent the need to learn goal-conditioned policies or value functions by combining IDMs with known symmetric properties of the robot. We train an IDM conditioned on the state space and a reduced goal space, using data collected while training any state-of-the-art RL algorithm. Our data collection is unique in that we exploit equivalences in experiences observed during training and learn a latent representation space shared between such equivalent experiences. Our IDM produces the action given this latent representation as an input, leading to generalization over parts of the state and goal spaces unobserved during training.

## 3 PRELIMINARIES

In the general RL framework (Sutton et al. (1998)), a learning agent interacts with an environment modeled as a Markov Decision Process consisting of: 1) a state space $\mathcal{S}$, 2) an action space $\mathcal{A}$, 3) a probability distribution $P: \mathcal{S} \times \mathcal{S} \times \mathcal{A} \to [0, 1]$, where $P(s'|s, a)$ is the probability of transitioning into state $s'$ by taking action $a$ in state $s$, 4) a reward function $\mathcal{R}: \mathcal{S} \times \mathcal{A} \times \mathcal{S} \to \mathbb{R}$ that gives the reward for this transition, and 5) a discount factor $\Gamma$. The agent learns a policy $\pi_\theta: \mathcal{S} \to \mathcal{A}$, parameterized by $\theta$ while trying to maximize the discounted expected return $J(\theta) = \mathbb{E}_{s_0, a_0, \ldots} [\sum_{t=0}^{\infty} \Gamma^t \mathcal{R}(s_t, a_t, s_{t+1})]$. Goal-conditioned RL optimizes for learning a policy that maximizes the return under a goal-specific reward function $\mathcal{R}_g$. On-policy RL algorithms, such as Policy Gradient methods (Williams (1992); Mnih et al. (2016)) and Trust Region methods (Schulman et al. (2015); Wu et al. (2017)) use deep neural networks to estimate policy gradients, or maximize a surrogate objective function subject to certain constraints. Off-policy RL algorithms (Lillicrap et al. (2016); Haarnoja et al. (2018)) incorporate elements of deep Q-learning (Mnih et al. (2013)) into the actor-critic formulation. Hindsight Experience Replay (HER) (Andrychowicz et al. (2017)) is a popular technique used in conjunction with an off-policy RL algorithm to learn policies in a sample-efficient way from sparse rewards in goal-based environments.

## 4 LEARNING GENERALIZED INVERSE DYNAMICS

Our method leverages samples collected while training a state-of-the-art RL algorithm to train an IDM that maps the current state and desired goal position to the action required to reach the goal. There are four major components involved in this process: 1) collecting data while training the RL algorithm, 2) learning a basic IDM that maps the current state and the desired goal to the required action, 3) collecting experiences that are equivalent to those observed, and using them to train the IDM, and 4) learning a latent representation that generalizes this model to unseen parts of the state space by utilizing the equivalent experiences collected in step 3. We elaborate on each of these in the following sections.

### 4.1 INITIAL TRAINING AND COLLECTING EXPERIENCE

Our goal in this steps is to collect data for our IDM in the process of learning a policy under a single reward function. Recall the motivation for learning IDMs: we want a model that can take in the current state of the agent and the next command, in the form of a location in the space that the

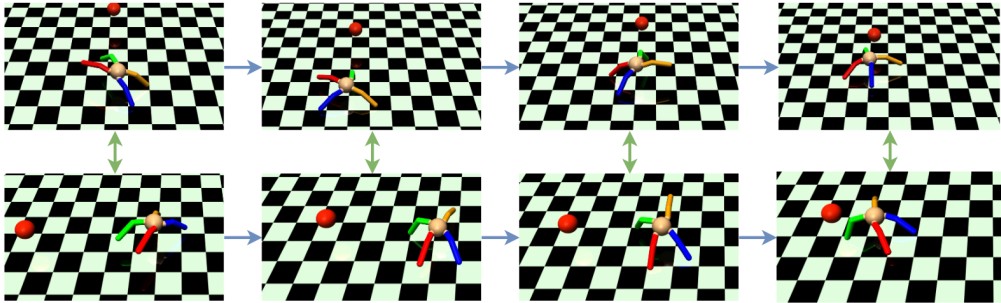

Figure 3: Example of trajectories that are equivalent modulo orientation (e.m.o.). The red sphere represents the goal. Here, we see that successively applying the same action to two Ants in different initial orientations results in trajectories that are e.m.o.

agent should travel to. Thus, we collect state, action, and position data from the transitions observed during the training process.

We emphasize the difference between the state space $S$ and the goal space $O$. The state space $S$ is high-dimensional, consisting of information related to joint angles, velocities, torques, etc. The goal space, $O$, is low-dimensional, consisting, in this case, of the 3-D coordinates of the goal position that the agent is supposed to navigate to.

**Definition 1** (Experiences). We define experiences as tuples $(s, o, o', a)$, where $s$ is the current state of the agent, $o$ is its current 3-D position, and $a$ is the action that the agent performed to move from the state $s$ and position $o$ to the next position, or intermediate goal, $o'$. We write $E_\tau = \{(s, o, o', a)_i\}_{i=1,...,T}$ to denote all the experience tuples collected from a trajectory $\tau$ of length $T$.

## 4.2 LEARNING THE INVERSE DYNAMICS MODEL

Given a set of experiences $E$, we can train the IDM using supervised learning techniques.

**Definition 2** (Inverse Dynamics Model). We define the Inverse Dynamics Model (IDM) as

$$\phi : \mathcal{S} \times \mathcal{O} \times \mathcal{O} \to \mathcal{A}, \ \phi(s, o, o') \to a \tag{1}$$

where $s$, $o$, $o'$ and $a$ represent the current state, current position, desired goal, and action respectively.

The IDM can reproduce seen actions on state and observation tuples that have appeared in the training data. However, it can not generalize good behaviour to states and observations that have not appeared in the initial training (see Fig. 6 for qualitative evidence). Our aim in the next steps is to generalize the observed experiences so that they can be used over previously unseen inputs from the $S \times O \times O$ space.

## 4.3 COLLECTING GENERALIZED EXPERIENCES

One issue with the current hypothesis of collecting data for a single reward is: the samples we obtain are highly biased in that they predominantly contain samples for motion in one direction. As a result, there are certain parts of the input space that our agent is unlikely to ever visit. So it is unreasonable to expect it to generalize its behaviour in those parts. We can see qualitative evidence in Fig. 6, where the Humanoid can navigate to goals seen during training time (6a) using the basic IDM (corresponding method in the plots is RL), but fails when the goal lies outside the training distribution (6b).

In order to mitigate this bias in the state space introduced by single reward function, we collect *generalized experience* comprising experience tuples that are *equivalent modulo orientation (e.m.o.)* with respect to actions. We use $\Theta$ to represent the orientation space.

**Definition 3** (Equivalence modulo Orientation). For $e_1, e_2 \in E$, $e_1$ and $e_2$ are e.m.o. with respect to $\mathcal{A} \implies \mathcal{A}(e_1) = \mathcal{A}(e_2)$. This defines an equivalence mod $\Theta$ with respect to $\mathcal{A}$ over $\mathcal{S} \times \mathcal{O} \times \mathcal{O}$

$$(s, o, o') \sim_\mathcal{A} (\hat{s}, \hat{o}, \hat{o}'), \text{ where } \forall a \in \mathcal{A}, p(o' \mid s, o, a) = p(\hat{o}' \mid \hat{s}, \hat{o}, a) \tag{2}$$

A qualitative example of e.m.o. experiences is shown in Fig. 3.

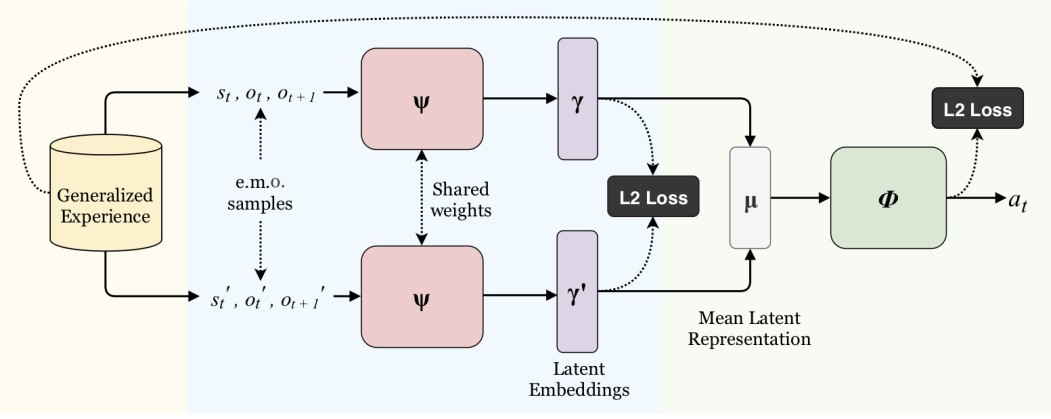

Figure 4: Learning latent representations using generalized experience: The LFM generates the latent representations for two e.m.o. samples. We minimize the distance between them to enforce equivalence, and also fit the IDM to predict the action correctly, given these latent representations.

**Definition 4** (Generalized Experience). We collect unseen experiences equivalent to the observed experiences by taking a trajectory $\tau \subseteq E$, changing the initial orientation of the agent, leading to a different $s_0$, and repeating the same set of actions observed in $\tau$. We denote this operation by $\mathcal{G}$, so $\mathcal{G}(\tau)$ is a new trajectory. The full generalized experience set obtained in this way is written as $G = E \cup \mathcal{G}(E)$. $s_0$ is an unseen state that has not appeared while training the agent.

### 4.4 LEARNING LATENT REPRESENTATIONS

Despite using generalized experiences during training, the IDM does not always show great improvements in tasks like navigating to the desired goal position in an arbitrary direction, as seen in Table 1. We hypothesize that this is due to the agent failing to recognize e.m.o. experiences, and instead learning actions from irrelevant attributes of the state space. We use the e.m.o. experiences from the generalized experience set $G$ to train a Latent Feature Model $\psi$ that discards irrelevant information from the state, and learns only shared information relevant to the IDM to produce actions.

**Definition 5** (Latent Feature Model). Our Latent Feature Model (LFM) aims to learn the equivalence between e.m.o. experiences from the generalized experience set $G$. We define the LFM $\psi$ as

$$\psi : G \to \mathbb{R}^k, \ \psi(s, o, o') = \gamma \tag{3}$$

where $\gamma$ is a $k$-dimensional latent representation of the experience sample. We then modify the IDM $\phi$ to produce the action from this latent representation as

$$\phi : \mathbb{R}^k \to \mathcal{A}, \ \phi(\psi(s, o, o')) = a, \forall (s, o, o') \in G \tag{4}$$

In order to learn these two models: LFM and IDM, we need our objective function to incorporate the property of equivalence modulo actions in the latent representations, and learn a good mapping from these representations to actions.

Since the LFM is used to generate latent representations for two e.m.o. experience samples simultaneously, and then optimize their distance, we use a Siamese framework (Koch et al. (2015)) to model $\psi$. Our objective $\mathcal{L}_1$ minimizes the distance between the latent representations generated for e.m.o. experience samples.

$$\mathcal{L}_1 = \|\psi(s, o, o') - \psi(\hat{s}, \hat{o}, \hat{o}')\|_2^2, \text{where } (s, o, o'), (\hat{s}, \hat{o}, \hat{o}') \in G, \text{and } (s, o, o') \sim_A (\hat{s}, \hat{o}, \hat{o}') \tag{5}$$

Next, we use a simple regression loss such as the L2 distance to fit the output of the IDM to the action. Here, the input to the IDM is the mean of the latent representations generated for the e.m.o. experiences.

$$\mathcal{L}_2 = \|\phi(\mu(\gamma, \hat{\gamma})) - a\|_2^2, \text{where } \gamma = \psi(s, o, o'), \hat{\gamma} = \psi(\hat{s}, \hat{o}, \hat{o}'), \text{and } (s, o, o') \sim_A (\hat{s}, \hat{o}, \hat{o}') \tag{6}$$

We jointly learn these two models by minimizing a weighted loss function:

$$\mathcal{L} = \lambda\mathcal{L}_1 + (1 - \lambda)\mathcal{L}_2, \text{where } \lambda \in (0, 1) \tag{7}$$

Fig. 4 shows the training procedure we use for our IDM. Each e.m.o. sample pair from equivalent trajectories is passed as input to the LFM, which generates the latent representations for the pair. The mean of these latent representations is then passed as input to the IDM, which predicts the action to be taken. These two models are trained simultaneously, resulting in rich latent representations that can achieve goal-directed motion, generalizable to any arbitrary location.

**Remark 1.** It is important to note that at test time, only the current state and goal are passed to the LFM to generate the latent representation, which is used by the IDM to predict the required action.

## 5 EXPERIMENTS

We demonstrate the effectiveness of our overall approach by performing a series of ablation experiments, successively including each major component of our algorithm. In addition to a random baseline, we use Vanilla Goal-Conditioned Policy, and Hindsight Experience Replay as baselines for comparison with our algorithm. We demonstrate superior results using our algorithm on three locomotion environments: Mujoco (Todorov et al. (2012)) Ant environment in OpenAI Gym (Brockman et al. (2016)), and Humanoid and Minitaur environments in PyBullet (Coumans & Bai (2016)). In order to fairly evaluate performance, we train each baseline using the same number of environment interactions as our algorithm. We also maintain uniform seeds, initial states and goals across all methods. The details of the test setting, along with network architectures, learning rates, and other hyperparameters, are discussed in the Appendix.

### 5.1 BASELINES

Since our method aims at achieving goal-directed motion, we compare it with other on-policy and off-policy RL algorithms that are trained for this specific purpose.

**Random Sampling (RS):** In this baseline experiment, we collect $(s, o, o', a)$ samples by taking random actions at each step. We use these samples to train the IDM.

**Vanilla Goal-Conditioned Policy (VGCP):** The second baseline algorithm we consider is VGCP, which takes as input the state and desired goal, and learns a policy on this input space using any state-of-the-art RL algorithm. The policy is given by $\pi_{\text{VGCP}} : \mathcal{S}, \mathcal{O} \to \mathcal{A}$ and is learnt using a state-of-the-art model-free RL technique. We use Proximal Policy Optimization (PPO) ((Schulman et al., 2017)) for Ant and Minitaur, and Soft Actor-Critic (SAC) ((Haarnoja et al., 2018)) for Humanoid.

**Hindsight Experience Replay (HER):** We select Soft Actor Critic (SAC) (Haarnoja et al. (2018)) and Deep Deterministic Policy Gradient (DDPG) (Lillicrap et al. (2016)) as the off-policy algorithms used in conjunction with HER for our algorithms. We report results on HER with both sparse and dense rewards. Sparse rewards indicate whether the target was successfully reached, and dense rewards include this information, in addition to control and contact costs. Throughout the paper, HER-Sp refers to HER with sparse rewards, and HER-De refers to HER with dense rewards.

### 5.2 ABLATION EXPERIMENTS

**Collecting Experience using standard RL algorithm (RL):** We collect samples while training state-of-the-art RL algorithms (Schulman et al. (2017); Haarnoja et al. (2018)) rewarding them for going to the right (as is common in locomotion environments), and use them as the training data for our IDM. More details are listed in the Appendix.

**Collecting Generalized Experience (GE):** We collect generalized experiences in the following manner: we save the trajectories followed by the agent while it learns a policy for locomotion. For some/all of these trajectories (details in A.1.2), we rotate the initial state of the agent by a random angle, and repeat the actions taken in the original trajectory. The samples collected in this modified trajectory are e.m.o. to those in the original trajectory. All of these samples constitute generalized experiences, which we use to train the IDM.

| Model | Distance to target | | Model | Distance to target | | Model | Distance to target | |
|---|---|---|---|---|---|---|---|---|
| | Mean | Std. dev. | | Mean | Std. dev. | | Mean | Std. dev. |
| RS | 3.159 | 0.579 | RS | 1.598 | 0.282 | RS | 3.166 | 0.569 |
| RL | 2.777 | 0.841 | RL | 1.403 | 0.438 | RL | 2.155 | 1.054 |
| VGCP | 2.160 | 0.955 | VGCP | 1.358 | 0.326 | VGCP | 2.608 | 0.607 |
| HER-Sp | 2.902 | 1.138 | HER-Sp | 1.440 | 0.237 | HER-Sp | 2.602 | 0.953 |
| HER-De | 2.891 | 1.135 | HER-De | 1.380 | 0.251 | HER-De | 2.475 | 0.693 |
| GE | 1.936 | 0.977 | GE | 1.212 | 0.485 | GE | 1.721 | 0.978 |
| **LR** | **1.779** | **0.986** | **LR** | **0.904** | **0.482** | **LR** | **1.105** | **0.866** |

Table 1: Quantitative results of our methods and all baselines on the three environments: Humanoid, Minitaur, and Ant respectively. In each environment, LR emerges as clearly the best performer, enabling the agent to navigate closest to the goal.

**Learning an IDM from Latent Representations (LR):** We use the generalized experiences collected in the previous step to extend the learned behaviour of the agent to unseen parts of the state space. We use the dual network architecture shown in Fig. 4 in this experiment. We jointly train the LFM and IDM using a weighted loss function that minimizes the distance between the latent representations generated for e.m.o. experiences, and fits the output of the IDM to the desired actions.

**Remark 2.** We preprocess $(s, o, o')$ to $(s, d)$ where $d$ is the unit vector in the direction $o \rightarrow o'$, and provide it as input to the models.

## 5.3 Navigating through a Series of Waypoints

We also show results on a task in which the agent has to navigate through a series of waypoints, to see if our IDM has indeed learnt the right actions. There are three questions we wish to answer through this experiment: 1) How much does the agent deviate from the intended optimal trajectory through the intermediate goals/waypoints? 2) How fast is the agent able to accomplish each goal? 3) What is the agent's style/gait of walking?

For this qualitative comparison, we found that neither HER nor VGCP with the current setting was able to generate good trajectories. Thus, we trained VGCP on 20 million samples (our method uses 15 million environment samples). We use this trained policy to generate the VGCP trajectories in Fig. 7, and our LR model (trained using 15 million samples) for the LR trajectories. For all trajectories, the agents are given the same initial orientation and goal position to ensure fair comparison.

## 6 Results

In this section, we analyze the performances of our algorithm and baselines qualitatively and quantitatively. In particular, we discuss the following: 1) overall performances of all methods, 2) distribution of distances to the goal observed at test time for all methods, 3) comparison between performance of best baseline and LR on the waypoint navigation task, and 4) analysis of speed, trajectory, and walking style observed in LR and the best baseline on Humanoid.

The first two points are addressed in the following two subsections. The next two questions are discussed in detail in Fig. 7.

## 6.1 Analyzing Overall Performance

We report the closest distance from the target that the agent is able to reach, as the evaluation metric for all our experiments. The mean and standard deviation of closest distance are reported in Table 1. It is clear that for each of the three environments, GE and LR outperform all other baselines. In particular, LR is observed to be the best performing algorithm for each environment. This lends validation to our claim that learning latent representations shared between equivalent experiences indeed boosts performance, instead of treating the equivalent experiences as independent training samples.

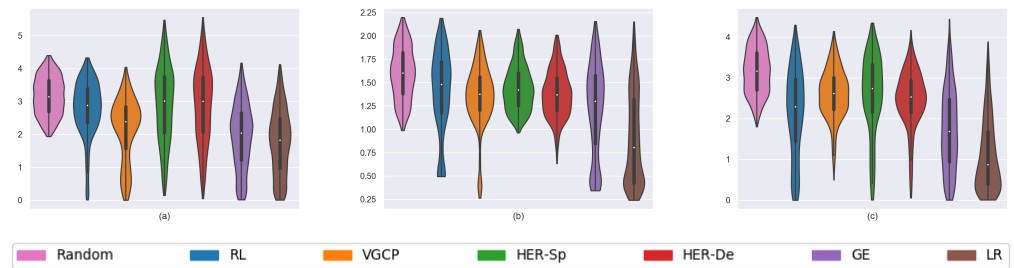

Figure 5: Violin plots showing the distribution of test data i.e. closest distance from the goal for each episode for the 3 environments: (a) Humanoid (b) Minitaur (c) Ant.

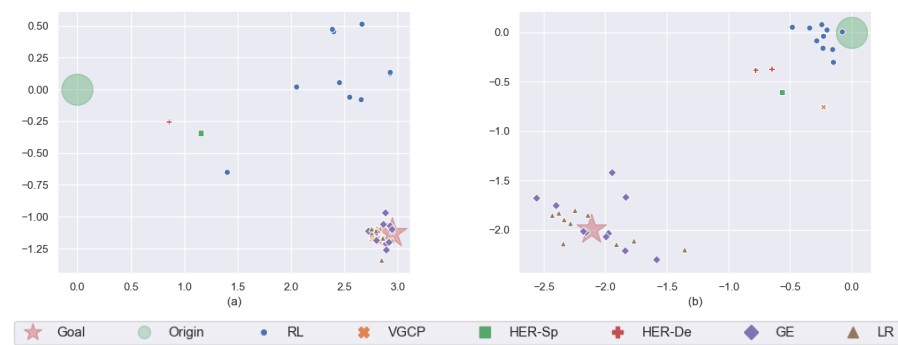

Figure 6: Qualitative comparison between different algorithms at test time for Humanoid.

The results in table 1 show that for the Minitaur and Humanoid environments, the performance of VGCP is better than most baselines, and nearly comparable to GE. However, it shows poor performance on the Ant environment. This anomaly arises from the fact that 2 million samples are not enough to train a goal-conditioned policy for the Ant. The Minitaur and Humanoid, on the other hand, are trained for 4 million and 15 million time steps respectively, thus enabling better goal-conditioned policies to be learnt, leading to much superior performance. For each of the environments, we observe that HER-Sp and HER-De both show poor performance, compared to that of GE, LR, and even VGCP.

## 6.2 DISTRIBUTION OF TEST SAMPLES

In this section, we analyze the violin plots in Fig. 5. These plots show the distributions of the closest distances from targets observed over 10,000 episodes for each algorithm. For each environment, we see that for RL, and in some cases, VGCP, HER, and GE, there is a small peak away from the mean, which gives a much lower distance than the mean distance. This suggests that there is a significant number of episodes in which the Humanoid reaches very close to the target. We analyze this discrepancy in performance qualitatively in 6, and conclude that the small peak consists of episodes in which the initial state and goal position have been observed during training, and thus, the goal-conditioned policy has already learnt the optimal actions for this scenario.

Across all environments, Ant and Minitaur in particular, the peak of the LR distribution is much lower than the other methods. This shows that in most episodes, the agent can successfully navigate to the goal; the slightly higher mean value and variance are due to the small number of episodes in which the agent fails at the beginning of the episode. This is to be expected because the actions taken by the IDM depend on the kind of data collected during the training process. In some cases, the model may learn the wrong actions, leading to the agent dying early in those episodes.

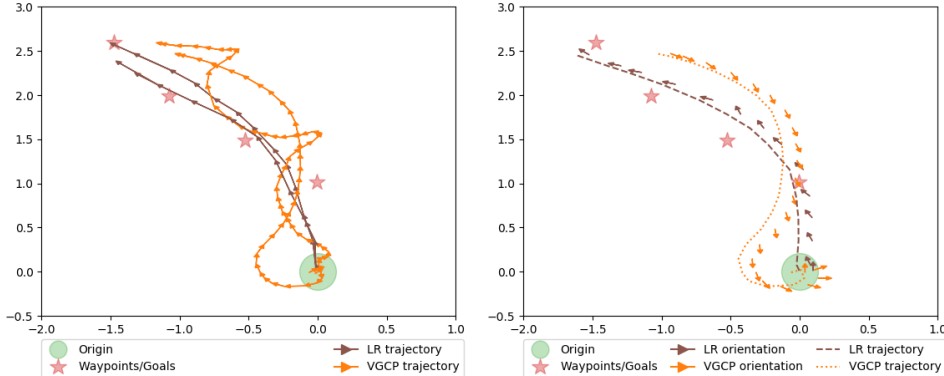

Figure 7: Given a series of waypoints, we check the ability of our Humanoid to navigate through them efficiently, and compare it with the best baseline, VGCP, trained on 20 million samples of environment interaction (our algorithm uses 15 million). (a) We plot the trajectory followed by each agent as it navigates through the waypoints. We see that LR can navigate through all goals much more easily than VGCP, without deviating. In addition to this, LR enables the agent to reach each goal much faster than VGCP, which is unable to reach the last goal due to its slow speed, as the episode terminates in a fixed number of timesteps. (b) We plot the orientation of the agent at uniform intervals throughout the episode. We observe that while the agent trained using LR walks forward as expected, the agent trained using VGCP, while managing to move slowly closer to the goal, does so in an arbitrary fashion in that its orientation is seldom in the direction of its motion.

We also provide qualitative evidence to prove that the distribution of closest distances in episodes is indeed biased by initial state and target configurations seen in the training data. We fix the initial state of the agent and generate goals in a specific region. We plot the results of 10 episodes for each method in Fig. 6 for the Humanoid environment. The initial state and goal configurations we show results for, consist of those experienced during training (left), and those not experienced during training (right). We see that for the configurations experienced at training time, most baseline methods, and our methods, GE and LR, are able to reach close to the goal. However, when the configuration lies outside the training distribution, only LR (sometimes GE) can navigate to the goal. We include more results in the Appendix.

# 7 CONCLUSION

We propose a new algorithm to achieve goal-directed motion for a variety of locomotion agents by learning the inverse dynamics model on shared latent representations for equivalent experiences. To this end, we take three important steps: (1) We utilize the experience collected by our agent while training a standard reinforcement learning algorithm, so that our IDM has "good" samples in which the agent walks reasonably well. (2) We generalize this experience by modifying the initial configuration for each observed trajectory in the collected data, and generate the equivalent trajectories. (3) We learn the important shared information between such symmetric pairs of experience samples through a latent representation that is used as an input to the IDM to produce the action required for the agent to reach the goal. We provide extensive qualitative and quantitative evidence to show that our methods surpass existing methods to achieve generalization over unseen parts of the state space.

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

## A APPENDIX

### A.1 EXPERIMENTAL SETUP

#### A.1.1 ENVIRONMENTS

We perform experiments on the Mujoco (Todorov et al., 2012) Ant environment in OpenAI Gym (Brockman et al., 2016), and Humanoid and Minitaur environments in PyBullet (Coumans & Bai, 2016). In each of these environments, for our methods, the agent is trained to perform well on the 3-D locomotion task in one direction. For the baseline methods, the agent is trained to reach goals generated in different 3-D positions.

The reward function for collecting data for RL, GE and LR includes contact and control costs, and rewards for moving forward. The reward function for training VGCP includes contact and control costs, but instead of moving forward, the agent is encouraged to move in the direction of the goal. In the case of HER-Sparse, the agent only receives a reward 0 if it reaches the goal, and -1 otherwise. For HER-Dense, the agent receives a weighted sum of distance to the goal and contact and control costs as its reward.

**Humanoid** Humanoid is a bipedal robot, with a 43-D state space and 17-D action space. The state contains information about the agent's height, orientation (yaw, pitch, roll), 3-D velocity, and the 3-D relative positions of each of its joints (knees, shoulders, etc.). The action consists of the torques at these joints.

**Minitaur** Minitaur is a quadrupedal robot, with a 17-D state space and 8-D action space. The state contains information about motor angles, torques, velocities, and the orientation of the base. The action consists of the torque to be applied at each joint.

**Ant** Ant is a quadrupedal robot, with a 111-D state space and 8-D action space. The state space contains the agent's height, 3-D linear and angular velocity, joint velocities, joint angles, and the external forces at each link. The action consists of the torques at all the 8 joints.

Our choice of these locomotion environments is driven by the motivation provided earlier: we want an agent to navigate to a desired goal position in the 3-D space. In view of this, we do not use 2-D locomotion environments like Half-Cheetah, Walker, or Hopper. Also, though Reacher and Pusher are goal-based environments, we do not use them as they are manipulation environments, and we are aiming to achieve navigation by agents trained on 3-D locomotion tasks.

### A.1.2 TRAINING DETAILS

The details of number of environment interactions, network architectures for the IDM, LFM, policy, and the state-of-the-art RL algorithm used to train the policy (for baselines), or collect data (for our methods), are given in Table 2.

We use the Adam optimizer (Kingma & Ba (2015)) with a learning rate of 1e-3 and a batch size of 512 for all our methods across all environments.

For the Humanoid environment, we observed that the first $\sim 5$ million samples were predominantly bad samples that led to the Humanoid falling down very early in each episode. Thus, we froze training after 5 million samples, when we had a policy in which the agent was able to walk a few steps. We used the policy at that point to collect 10 million steps of environment interaction. For the Ant and Minitaur environments, we collected 2 million and 4 million steps of environment interactions respectively, during training.

For RL, we used all steps encountered in on-policy data. For GE and LR, we used only the first half of training samples encountered while training the policy. The second half of the samples is generated by applying the generalized transformation $\mathcal{G}$, explained in Section 4.3. We use the first 1 million, 2 million and 5 million samples collected for the Ant, Minitaur and Humanoid (trained) respectively, and generate the rest by applying $\mathcal{G}$.

The hyperparameters involved in training the LFM and IDM are: the dimension $k$ of the latent representation space, and the ratio of the latent representation loss ($\mathcal{L}_1$) to the regression loss ($\mathcal{L}_2$), $\lambda$. We set $\lambda = 0.25$ for all 3 environments. We use $k = 10$ for the Ant and Minitaur environments, and $k = 50$ for the Humanoid environment. We discuss the impact of the latent representation dimension $k$ on the performance of the model in the next section.

### A.1.3 TEST SETTING

At test time, the agent is rotated by a random angle. The target is set at a distance of $\sim 2 - 5$ units from the agent at an angle of $[-45°, 45°]$ to the agent's orientation in the case of the Ant and Humanoid environments. For Minitaur, we set the target at a distance of $\sim 1.5 - 2.5$ units from the agent at an angle of $[-45°, 45°]$ to the agent's orientation. The intermediate goals, or waypoints, can be provided by any planning algorithm, since our approach is agnostic to the actual planning algorithm used. In our work, we use Model Predictive Control, i.e. we replan at each step.

Each episode consists of a maximum of 1000 steps for each environment, and the episode terminates when the agent reaches the goal, or falls down/dies. We report the closest distance from the target that the agent is able to reach, for each episode. We select 10 random seeds and test the performance of each method on 1000 episodes for each random seed. In order to ensure that the comparison between all methods is indeed fair, we set the initial configuration of the agent and the target to be the same across all methods at test time.

### A.2 ADDITIONAL ANALYSIS OF RESULTS

**Why does our method perform better than baselines in spite of using inferior samples?** It is important to note that since we take only the first half of the environment interactions for GE and LR, we are learning from essentially inferior samples consisting of actions that may not be optimal, as they have been encountered earlier in the training process. In spite of learning from inferior actions, our method outperforms the baselines. This is because our IDM formulation enables the learning of actions that can reach the next goal given the current state, which is the kind of behaviour we require in goal-directed motion. The focus of this model is more on learning actions that caused certain transitions, not on learning the most optimal actions that achieve the transition, although that would be the next best thing to do. In addition to this, our LFM framework enables the agent to achieve their equivalent transitions by producing the same action for all transitions that share a common latent representation. This enables our agent to generalize its motion to a larger part of the goal space in comparison with baselines that struggle to achieve goal-directed motion for goals or states lying outside the training distribution.

| Environment | # of Interactions | Method | Architecture | Training Algorithm |
|---|---|---|---|---|
| Humanoid | 15M | RS | IDM: $256 \times 256$ | - |
| | | RL | Policy: $256 \times 256$
IDM: $256 \times 256$ | SAC |
| | | VGCP | Policy: $256 \times 256$ | SAC |
| | | HER-Sparse | Policy: $64 \times 64$ | SAC |
| | | HER-Dense | Policy: $64 \times 64$ | SAC |
| | | GE | Policy: $256 \times 256$
IDM: $256 \times 256$ | SAC |
| | | LR | Policy: $256 \times 256$
LFM: $256 \times 256$
IDM: $256 \times 256$ | SAC |
| Minitaur | 4M | RS | IDM: $256 \times 256$ | - |
| | | RL | Policy: $256 \times 256$
IDM: $256 \times 256$ | PPO |
| | | VGCP | Policy: $256 \times 256$ | PPO |
| | | HER-Sparse | Policy: $64 \times 64$ | SAC |
| | | HER-Dense | Policy: $64 \times 64$ | SAC |
| | | GE | Policy: $256 \times 256$
IDM: $256 \times 256$ | PPO |
| | | LR | Policy: $256 \times 256$
LFM: $256 \times 256$
IDM: $50 \times 50$ | PPO |
| Ant | 2M | RS | IDM: $256 \times 256$ | - |
| | | RL | Policy: $64 \times 64$
IDM: $256 \times 256$ | PPO |
| | | VGCP | Policy: $64 \times 64$ | PPO |
| | | HER-Sparse | Policy: $64 \times 64$ | SAC |
| | | HER-Dense | Policy: $64 \times 64$ | DDPG |
| | | GE | Policy: $64 \times 64$
IDM: $256 \times 256$ | PPO |
| | | LR | Policy: $64 \times 64$
LFM: $256 \times 256$
IDM: $50 \times 50$ | PPO |

Table 2: Training details for Humanoid, Minitaur, Ant. Note that training algorithm refers to the RL algorithm used to collect data (for our methods), or train the policy (for baselines). Note that the hyperparameters for all experiments (VGCP, HER, PPO, SAC), have been taken from existing implementations of these algorithms (Dhariwal et al. (2017); Hill et al. (2018)).

**Why is our method more sample-efficient than the baselines?** Our methods (GE and LR) do not utilize the latter half of samples collected during the training process, which would consist of more optimal actions, because that would imply using a higher number of actual environment interactions than the baselines. In spite of this, our method outperforms all baselines for all environments. This is because goal-conditioned policies explore the state and goal spaces to achieve goal-directed motion. While exploration is in fact a desirable component in reinforcement learning, sampling from the goal space during training requires the agent to acquire a sense of direction and the skill of locomotion simultaneously. This enlarged state space, comprising the original state and the goal, leads to an increase in the number of samples required by the agent to learn a good policy. Our latent representations, on the other hand, encode the property of equivalence modulo orientation with respect to actions, described in Section 4.3. This enables our method to generalize representations of some samples of training experience to other parts of the state and goal spaces, which have not been encountered during training. As a result, the IDM learns to predict actions for states and goals it has never encountered before, if they have the same latent representation as those seen during training, thus boosting performance while remaining sample-efficient.

**Why does LR show qualitatively better results than VGCP?** VGCP is trained as a function of the state and goal space, and produces the action that the agent should take in order to reach the goal. The agent is rewarded for moving towards the goal, keeping control and contact costs minimal. There is no constraint on the type of motion exhibited by the agent, or the time taken by the agent to reach the goal. As a result, we see in Fig. 7 that the agent trained using VGCP walks very slowly throughout both episodes, and follows a non-optimal path to navigate through 3 out of the 4 waypoints. Due to its slow speed, it is unable to reach the last waypoint. The LR agent, on the other hand, learns an IDM from samples collected while training on the locomotion task. Since the locomotion reward encourages walking forward while also keeping control and contact costs minimal, the LR agent always walks forward facing each waypoint, and follows a smooth trajectory through all the waypoints, validating the superior performance of our IDM. We see that though the LR and VGCP agents are initialized in the same orientation, the VGCP agent walks with its back facing the waypoints, while the LR agent successfully adopts the optimal path and navigates through all waypoints adopting a "natural" walking style.

**Why does HER not perform as well as other baselines?** If we revisit the motivation behind the HER algorithm, we realize that it shows superior performance using fewer training samples over environments that have low-dimensional state and action spaces, and a sparse reward setting. In our case, we are dealing with high-dimensional locomotion agents that require specific actions to walk, and navigate to the goal. If we use the sparse reward setting, most samples in the replay buffer consist of failures, as the agent first needs to learn to walk, after which it can successfully navigate to the goal. Even with the dense reward setting, there is no significant improvement in performance because of this reason. HER performs very well on low-dimensional sparse-reward environments but it is difficult to extend this behaviour to higher dimensions or dense rewards, especially where complex locomotion skills have to be learned in order to navigate to the goal.

**How important is the latent representation dimension $k$ in achieving good performance?** The dimension of the latent representation $k$ is perhaps the most important hyperparameter that determines the performance of LR. Choosing a bad value of $k$ could result in information loss if $k$ dimensions are not sufficient to encode the relevant information for producing the action, or irrelevant information being encoded if $k$ is too high. In our experiments, we use $k = 10$ for Ant and Minitaur, and $k = 50$ for Humanoid. We show the impact of $k$ on performance in the Ant environment in Fig. 9. We see that performance improves as we keep reducing the dimension up until $k = 10$. As we try to reduce further, performance suddenly drops, implying loss of information. Thus, the Ant's 111-D state and 3-D goal are reduced to a 10-D representation that is used by the IDM to produce actions to navigate to the goal.

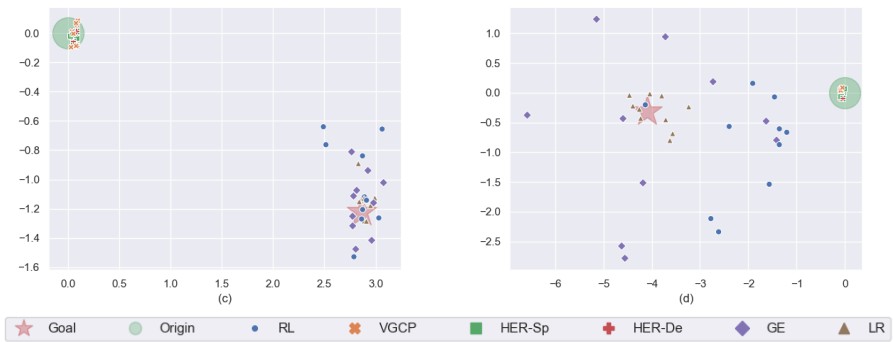

Figure 8: Qualitative comparison between different algorithms at test time for Ant.

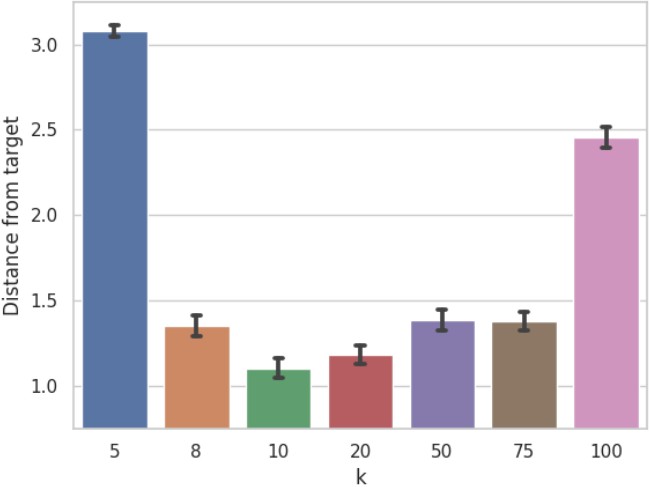

Figure 9: Impact of latent representation $k$ on performance for the Ant environment

