# OpenReview forum: "Learning Latent Representations for Inverse Dynamics using Generalized Experiences"
_ICLR.cc/2020/Conference — Reject_

### Official Review · AnonReviewer1 · 2019-10-19
**Official Blind Review #1**

**Rating:** 3

**Review:**

Learning Latent Representations for Inverse Dynamics using Generalized Experiences

In this paper, the authors propose to utilize the symmetry property in locomotion problems (more specifically navigation problems), and more efficiently generate additional training data from existing data, and learns a more efficient representation.

I tend to vote for rejection for this paper mostly because, while it seems to me to be a very efficient and practical engineering project, but relatively lack the novelty in terms of the algorithm.

Pros:
- The experiments are of good quality, providing a lot of ablation studied and hyper-parameter specifications.
- The proposed idea is combined with some of the state-of-the-art algorithms, showing it’s compatibility and good practical performance.

Cons:

- The proposed algorithm lack novelty.
Goal conditioned reinforcement learning, where a generalized inverse dynamics is used, has been widely studied in [2, 3].
And the use of symmetry has also been studied [1].
The augmentation of data by considering symmetry is relatively straight-forward.

- limited to navigation environments
The proposed methods do not seem to be directly applicable to tasks other than navigation, where a very task-specific goal position can be provided.


[1] Yu, Wenhao, Greg Turk, and C. Karen Liu. "Learning symmetric and low-energy locomotion." ACM Transactions on Graphics (TOG) 37.4 (2018): 144.
[2] Ding, Yiming, Carlos Florensa, Mariano Phielipp, and Pieter Abbeel. "Goal-conditioned Imitation Learning." arXiv preprint arXiv:1906.05838 (2019).
[3] Merel, Josh, Leonard Hasenclever, Alexandre Galashov, Arun Ahuja, Vu Pham, Greg Wayne, Yee Whye Teh, and Nicolas Heess. "Neural probabilistic motor primitives for humanoid control." arXiv preprint arXiv:1811.11711 (2018).

**Experience Assessment:**

I have published in this field for several years.

**Review Assessment: Checking Correctness Of Derivations And Theory:**

I carefully checked the derivations and theory.

**Review Assessment: Checking Correctness Of Experiments:**

I carefully checked the experiments.

**Review Assessment: Thoroughness In Paper Reading:**

I read the paper thoroughly.

---

> ### Author Response · Authors · 2019-11-08
> **Author Response to Reviewer #1 - Point 1**
>
> We thank the reviewer for the constructive feedback. We hope that our revision and the following comments answer the questions raised:
>
> "The proposed algorithm lack novelty. Goal conditioned reinforcement learning, where a generalized inverse dynamics is used, has been widely studied in [2, 3]."
>
> Goal-conditioned reinforcement learning involves maximizing the return under a goal-specific reward function. Our work, in contrast, leverages experience already collected while training an agent for a locomotion task, to enable the agent to navigate to the goal. Our approach is different in that we do not introduce the goal while training the locomotion policy. We train an agent to perform well on the locomotion task, and then utilize the data collected in this process to learn a model (IDM) that maps the current state of the agent and the goal, to the required action. Thus, our method is different from goal-conditioned reinforcement learning methods.
> [2] develops an algorithm for goal-conditioned RL from the imitation learning perspective, i.e. a set of expert trajectories is provided to the learning agent, whereas our method does not have an expert demonstrator. In addition to this, the data augmentation technique used in [2] leverages intermediate states in the expert trajectory as goals, whereas our method leverages symmetry in the trajectories collected by the agent while optimizing for a locomotion policy.
> [3] has a very different setting from our work: a large number of experts that perform single skills well are available, and the focus is on learning a shared policy that can perform well on single skills, or a composition of skills, to eventually achieve a system capable of performing one-shot imitation. Also, the embedding space learned in [3] represents short-term motor behaviour, and is used to pick behavioral modes, whereas the embedding space we learn, captures orientation-invariant features that help the agent generalize its actions for the state-goal input space.
> The main challenge in existing approaches for goal-conditioned RL is the need to sample goals, resulting in low generalization capability (as we show in VGCP, HER-Sp and HER-De). Thus, we propose learning latent representations for encoding information shared between similar experiences, and learn an inverse dynamics model that takes as input these latent representations and outputs the desired actions. We compare our results (GE, LR), with methods that use goal-conditioned policies/value functions (VGCP, HER-Sp, HER-De), and show that our method is much superior both qualitatively, and quantitatively.
>
>
> "And the use of symmetry has also been studied [1]."
>
> It is true that symmetry has been studied in [1]; however, [1] imposes symmetry in the action space of the locomotion agent, by penalizing paired limbs for learning different actions, to learn good locomotion gaits. Our work utilizes symmetry in state-goal pairs in different orientations, to achieve navigation to arbitrary goals in a sample-efficient manner.
>
>
> "The augmentation of data by considering symmetry is relatively straight-forward."
>
> We have shown that merely augmenting the data by adding symmetric experiences is not enough to achieve SOTA performance (refer to GE results in Table 1). It is essential to learn a common embedding that is invariant to orientation, and can control the agent effectively in any orientation to reach the goal (refer to LR in experiments). The results of our ablation studies clearly indicate that LR (architecture shown in Fig. 4) outperforms GE (using generalized experiences to train the IDM) in each environment, showing that the design of the Latent Feature Model with shared weights for e.m.o. experiences leads to a boost in performance over using only symmetric experiences, for all agents.
>
>
>
> [1] Yu, Wenhao, Greg Turk, and C. Karen Liu. "Learning symmetric and low-energy locomotion." ACM Transactions on Graphics (TOG) 37.4 (2018): 144.
> [2] Ding, Yiming, Carlos Florensa, Mariano Phielipp, and Pieter Abbeel. "Goal-conditioned Imitation Learning." arXiv preprint arXiv:1906.05838 (2019).
> [3] Merel, Josh, Leonard Hasenclever, Alexandre Galashov, Arun Ahuja, Vu Pham, Greg Wayne, Yee Whye Teh, and Nicolas Heess. "Neural probabilistic motor primitives for humanoid control." arXiv preprint arXiv:1811.11711 (2018).

---

> > ### Author Response · Authors · 2019-11-08
> > **Author Response to Reviewer #1 - Point 2**
> >
> > "limited to navigation environments - The proposed methods do not seem to be directly applicable to tasks other than navigation, where a very task-specific goal position can be provided."
> >
> > We have shown results on tasks that have been the focus of many recent works in goal-conditioned RL ([4], [5], [6]). At the same time, the key definition of e.m.o. experiences is not restricted to this setting, because they can be used whenever symmetry in data can be exploited.
> >
> >
> >
> > [4] Vitchyr Pong*, Shixiang Gu*, Murtaza Dalal, and Sergey Levine. Temporal difference models: Model-free deep RL for model-based control. In International Conference on Learning Representations, 2018.
> > [5] Dibya Ghosh, Abhishek Gupta, and Sergey Levine. Learning actionable representations with goalconditioned policies. CoRR, abs/1811.07819, 2018.
> > [6] Florensa, C., Held, D., Geng, X. and Abbeel, P., 2017. Automatic goal generation for reinforcement learning agents. arXiv preprint arXiv:1705.06366.

---

### Official Review · AnonReviewer3 · 2019-10-21
**Official Blind Review #3**

**Rating:** 3

**Review:**

This paper proposes a method to learn locomotion and navigation to a goal location or through a set of waypoints for simulated legged robots. The contributions of this paper include 1) generalized experience, which is a data-augmentation technique to add more orientation-invariant experience, and 2) a latent representation to encode the state, the current location and the goal location. The paper compares the proposed method with a few baselines and demonstrates better performance.

My recommendation of this paper is Weak Reject. Although the method seems reasonable and the evaluation shows good results, I think that the paper can be improved for the following three reasons.

First, it is not clear to me how the inference works if the goal is reasonably far away (e.g. can be reached in 10 steps) from the current position of the robot? Since the inverse dynamics model only outputs an action given the current position and the immediate goal (in the next time-step), how is the 10-step action sequence planned using the 1-step immediate goals?

Second, the details of data collection are unclear to me. I believe that the policy \pi used for data collection plays an important role. Which \pi is used? It would be clearer to present this part in the main text, not Appendix. If the data is collected from an RL agent which learns to walk to the right, how does the robot learns to turn when walking across different waypoints (Figure 7)?

Third, in this paper, the generalized experience is to add different initial orientations of the robot. I think that the similar effect can be achieved by reparameterizing (s, o, o') into polar coordinates: (s, \theta, r), where (\theta, r) is the intermediate goal location relative to the robot's current orientation and position. \theta=0 means the goal is in front of the robot, and r is the distance between the goal and the robot. In this representation, all the generalized experience will reduce to a single (s, \theta, r), which is invariant to the robot's orientation. This would already be a good latent space, without any learning. For this reason, I would suggest adding one more baseline to compare the proposed method against: using (s_t, \theta_t, r_t, a_t, s_{t+1}, \theta_{t+1}, r_{t+1}) to represent experience, then run the proposed method without generalized experience and latent representation. Will this baseline achieve similar or even better results?

**Experience Assessment:**

I have published one or two papers in this area.

**Review Assessment: Checking Correctness Of Derivations And Theory:**

I assessed the sensibility of the derivations and theory.

**Review Assessment: Checking Correctness Of Experiments:**

I assessed the sensibility of the experiments.

**Review Assessment: Thoroughness In Paper Reading:**

I read the paper at least twice and used my best judgement in assessing the paper.

---

> ### Author Response · Authors · 2019-11-08
> **Author Response to Reviewer #3**
>
> We thank the reviewer for the constructive feedback. We hope that our revision and the following comments answer the questions raised:
>
> 1. The setting of our IDM is that it takes in waypoints, or intermediate goals, and outputs the action required to reach that goal. These waypoints are typically provided by a planning algorithm. Our approach is agnostic to the planning algorithm; in our methods, we use Model Predictive Control, i.e. we replan at each step. We have added these details to A.1.3.
>
>
> 2. Thank you for pointing this out. We have modified the paragraph on “Collecting Generalized Experiences (GE) in 5.2 to make the data collection process clearer. In short, data collection for our methods happens in the following manner:
> a) An agent learns to walk to the right. We collect the trajectories observed during this training procedure.
> b) For each trajectory (this varies according to the environment, exact details in A.1.2), we rotate the initial state of the agent by a random angle, and repeat the actions taken in the trajectory.
> c) These two trajectories (the original one, and the augmented one), consist of e.m.o. experience samples (Def. 3). These samples are used to train our models (GE and LR).
>
> The reason that the agent is able to turn while walking across different waypoints as seen in Fig. 7 is that thought the agent is rewarded for walking to the right, there is no penalty on it drifting from the X-axis. This diversity of samples benefits the robustness of the learned IDM. Thus, the agent will encounter a number of trajectories, in which it is not strictly going to the right, but along a curve, and still gets high rewards. These kinds of experience samples allow our IDM to learn actions that can enable the agent to turn in a certain direction, resulting in the trajectories that you see in Fig. 7.
> For the baseline algorithm VGCP used in Fig. 7, at training time, goals are randomly sampled from all directions, and so, the agent is able to learn policies to navigate to any goal in the plane.
>
>
> 3. We have updated the main text to include the exact input to our models (please refer to Remark 2). Instead of providing (s, o, o’) to the model, we input (s, d), where d is the unit vector in the direction of the goal (represents \theta). Since we replan at each step, we believe we do not need to include the distance to the goal as an input to the model. Thus, the experiment requested (running the proposed method without generalized experience and latent representation) is the same as RL in the paper (see 5.2).

---

### Official Review · AnonReviewer2 · 2019-10-27
**Official Blind Review #2**

**Rating:** 3

**Review:**

The paper proposes a method for exploiting structure in locomotive tasks for efficiently learning low-level control policies that pass through waypoints while achieving some goal (typically 3D Cartesian position). This is in contrast to goal-conditioned RL policies that sample random goals during training and are thus sample inefficient, which are trained to execute one policy at a time. In particular, the paper proposes the notion of generalized experiences, where new trajectories are generated from existing trajectories, in such as a way that they are equivalent to each other (in this case translation and orientation invariant) with respect to actions.

The idea proposed here, of exploiting environmental structures in order better generalize previous knowledge to new, unseen situations is an interesting direction for achieving sample efficiency in practical RL, such as in robotics.

I have the following comments/questions.

1. If I understand correctly, rather than randomly sampling the environment, the paper proposes starting off with trajectories generated while learning some single-goal policy, and generate from these new ones that the agent must execute in the environment. In which case, the question is how much less interaction does the agent have with the environment compared to random goal sampling, to achieve the same performance?

2. What happens if you use a standard goal-conditioned RL with the Generalized Experiences, without training an IDM? For example, using VGCP with GE, where the goals for VGCP are teminating states of each trajectory. In other words, can a vanilla goal-conditioned RL benefit from the proposed trajectory sampling technique (GE), and how does that compare with random sampling and using the proposed latent representation technique?

3. How do you modify the LFM so that it only accepts the current state and goal, instead of a set of two e.m.o equivalent states and goals? Is the same query treated as two queries that are e.m.o equivalent?

4. In A.1.2, the paper mentions that the first half of the data generated using the RL algorithm is used for generating the second half of data for GE and LR, right? Which means GE and LR are trained from the same number of steps as the baselines, except that GE and LR make use of generalized experiences. However, the paper states in A2 that GE and LR learn from inferior samples. Can the authors please clarify this? My understanding is that GE are generated by modifying existing trajectories and letting the agent apply the same actions by interating with environment. Meaning that, given 2M samples, GE will generate 2M more samples that are e.m.o equivalent by interacting with the environment.

5. Some of the references in the text do not have year of publication, e.g., Kulkarni et al.

**Experience Assessment:**

I have read many papers in this area.

**Review Assessment: Checking Correctness Of Derivations And Theory:**

I carefully checked the derivations and theory.

**Review Assessment: Checking Correctness Of Experiments:**

I carefully checked the experiments.

**Review Assessment: Thoroughness In Paper Reading:**

I read the paper thoroughly.

---

> ### Author Response · Authors · 2019-11-08
> **Author Response to Reviewer #2**
>
> We thank the reviewer for the detailed feedback. We hope that our revision and the following comments answer the questions raised:
>
> 1. For the Mujoco Ant environment, our results show that naive sampling of the goal space gives models that are still far from matching the performance shown by our models, after 10 million samples of environment interaction (our models use only 2 million samples). Thus, we have focused on comparing with more sophisticated baseline methods as shown in Section 5. But this is a good point and we can add more details to the paper.
> Intuitively, we can think about this as: if (s1, g1) and (s2, g2) have the same angle between the agent’s orientation and the goal, a policy learned using goal-conditioned RL could learn different actions to reach the goal in each case, which would be more expensive in terms of the number of samples used to learn these actions. However, in our case, by modeling the equivalence between these (state, goal) pairs into a common latent embedding, we are generalizing the same actions across all such pairs. Thus, our method would be more sample-efficient than random goal sampling.
>
>
> 2. Thank you for raising this point. Although it is not the focus of our paper, adding generalized experiences would likely help learning the goal-conditioned policy as well. The setting in our paper assumes that the IDM takes pairs of waypoints and outputs actions, where the waypoints can be given by a higher level planning module. This is the typical setting in locomotion robots, but using generalized experiences to directly handle goal-conditioned RL by combining the two modules is an interesting avenue for future work.
> The point discussed in (1) still holds in this case: using these generalized experiences, the goal-conditioned policies may enable the agent to navigate to a larger range of goals, but it would not be generalizable like our methods, since it does not attempt to learn the symmetries in these experiences.
>
>
> 3. We do not modify the LFM in any way. The set of 2 e.m.o. states and goals in Fig. 4 share weights i.e. they are passed through the same neural network to generate the latent representations. As mentioned in Remark 1 (page 6), at test time, only the current state and goal are passed through this network to generate the latent representation.
>
>
> 4. Although the same number of samples is used for training all models, GE and LR use the first half of the samples collected while training the agent to walk to the right. These samples are “inferior” to the latter half of samples, as the agent achieves lower rewards in the first half. In other words, as the agent interacts more with the environment and updates its policy, it learns better locomotion policies. In this sense, the samples used to train GE and LR are collected by running an inferior locomotion policy, compared to the other models.
>
>
> 5. Thank you for pointing this out. We have corrected these references in the revision.

---

### Decision · Program_Chairs · 2019-12-19

**Decision:**

Reject

**Comment:**

Solid, but not novel enough to merit publication.  The reviewers agree on rejection, and despite authors' adaptation, the paper requires more work and broader experimentation for publication.